# Temporal dynamics of animacy categorization in the brain of patients with mild cognitive impairment

Hamed Karimi [1,2]*, Haniyeh Marefat[3], Mahdiyeh Khanbagi[1], Chris Kalafatis[4,5,6], Mohammad Hadi Modarres[6], Zahra Vahabi[7,8], Seyed-Mahdi Khaligh-Razavi[1,6]*

1 Department of Stem Cells and Developmental Biology, Cell Science Research Center, Royan Institute for Stem Cell Biology and Technology, ACECR, Tehran, Iran, 2 Department of Mathematics and Computer Science, Amirkabir University of Technology, Tehran, Iran, 3 School of Cognitive Sciences, Institute for Research in Fundamental Sciences (IPM), Tehran, Iran, 4 South London & Maudsley NHS Foundation Trust, London, United Kingdom, 5 Department of Old Age Psychiatry, King's College London, London, United Kingdom, 6 Cognetivity Ltd, London, United Kingdom, 7 Department of Geriatric Medicine, Ziaeian Hospital, Tehran University of Medical Sciences, Tehran, Iran, 8 Memory and Behavioral Neurology Division, Roozbeh Hospital, Tehran University of Medical Sciences, Tehran, Iran

* hamedk72@gmail.com (HK); seyed@cognetivity.com (SMKR)

**Data Availability Statement:** The data for the behavioral tests (MoCA, ACE-R, and ICA) can be found in the supplementary materials (S1 Table). The EEG dataset related to the findings in the

## Abstract

Electroencephalography (EEG) has been commonly used to measure brain alterations in Alzheimer's Disease (AD). However, reported changes are limited to those obtained from using univariate measures, including activation level and frequency bands. To look beyond the activation level, we used multivariate pattern analysis (MVPA) to extract patterns of information from EEG responses to images in an animacy categorization task. Comparing healthy controls (HC) with patients with mild cognitive impairment (MCI), we found that the neural speed of animacy information processing is decreased in MCI patients. Moreover, we found critical time-points during which the representational pattern of animacy for MCI patients was significantly discriminable from that of HC, while the activation level remained unchanged. Together, these results suggest that the speed and pattern of animacy information processing provide clinically useful information as a potential biomarker for detecting early changes in MCI and AD patients.

## 1 Introduction

Mild Cognitive Impairment (MCI) is a condition in which an individual has a mild but measurable decline in cognitive abilities. This decline is noticeable to the person affected and to the family members and friends, but the individual can still carry out everyday activities [1, 2]. A systematic review of 32 cohort studies shows an average of 32 percent conversion from MCI to Alzheimer's Disease (AD) within a five-year follow-up [3]. 5–15% of people with MCI have also been shown to develop dementia every year [4].

Electroencephalography (EEG) is widely used to study the resting-state neural activity in the brain of patients with MCI and mild AD [5–14]. A few studies have also used EEG to

presented manuscript is available at RepOD (https://doi.org/10.18150/DEQMGF).

**Funding:** This work was partly supported by the Iranian Cognitive Sciences, and Technologies Council's (COGC) grant (#4873) awarded to SKR. Cognetivity ltd. provided support in the form of salaries for authors SKR, CK, MHM. The funders did not have any additional role in the study design, data collection, analysis, decision to publish, or preparation of the manuscript.

**Competing interests:** SKR and CK serve as the CSO and CMO at Cognetivity ltd., and HModarres as the lead data scientist. These affiliations do not alter our adherence to PLOS ONE policies on sharing data and materials. Other authors declared no competing interests.

associate abnormalities in memory function with cognitive impairment during both encoding and decoding stages of working memory [7, 15–17] as well as episodic memory tasks [18].

The relationship between cognitive impairment and visual system changes has recently gained attention [19]. Several studies have linked deficiencies in different parts of the visual system with AD [20–22]. There are several documented cases in which visual function problems are the initial and dominant manifestation of dementia [23, 24]. A few studies have also used a visual task to report changes in the EEG responses of patients with MCI and AD [25, 26].

Several studies of the visual system in primates and healthy human subjects have demonstrated that images are categorized by their animacy status (i.e., animate vs. inanimate) in the higher-level visual areas, i.e., inferior temporal (IT) cortex [27–32]. The neural activity underlying the animacy information processing of briefly flashed images was also studied in healthy adults [33, 34]. Studies have shown that animacy information emerges in the brain of healthy human subjects as early as 80±20 ms after the stimulus onset and reaches its peak within 250 ±50ms after the stimulus onset [35–37]. However, the underlying neural dynamics of animacy processing in the brain of patients with MCI in comparison to healthy controls (HC) is still unknown. Several studies using autopsy [38–41] and PET imaging [42, 43] have shown that some of the visual areas in the temporal cortex are among the first regions affected by the disease.

The Integrated Cognitive Assessment (ICA) is a visual task based on a rapid categorization of natural images of animals and non-animals [44]. ICA assesses changes in the speed and accuracy of animacy processing in patients with MCI and mild AD and is shown to be sensitive in detecting MCI patients [45, 46]. The ICA primarily tests Information processing speed (IPS) and engages higher-level visual areas in the brain for semantic processing, i.e., distinguishing animal vs. non-animal images [44]. Reduced visual processing speed is reported in amnestic-MCI [47], and IPS is further reported to be related to other areas of cognitive dysfunction [48, 49].

In line with previous behavioral studies, we hypothesized that the underlying neural response of MCI patients during an animacy categorization task is both slower and less accurate at the level of representation compared to HC [44]. The absence of multivariate methods in previous studies also raises the question of whether the pattern of animacy information processing is different in patients with MCI compared to that of HC.

To address these questions, we acquired EEG data from MCI and HC participants during the completion of the ICA's animal/non-animal categorization task. We studied the temporal neural dynamics of animacy processing in MCI patients using both univariate and multivariate analyses. By applying multivariate pattern analysis (MVPA), we compared the neural speed of animacy processing in MCI and HC. We further looked beyond the conventional univariate methods and compared MCI and HC in terms of their pattern of EEG responses to natural images of animal and non-animal stimuli.

We found that the categorical representation of animacy information emerges later in the brain of patients with MCI compared to that of HC. Furthermore, the results reveal differences between the EEG response patterns of HC vs. MCI during the time-points when univariate mean responses showed no significant difference. The EEG response patterns could further be used to discriminate HC from MCI, demonstrating that the pattern of EEG activity also carries information about the status of the disease beyond the conventional univariate analysis of mean activities.

## 2 Methods

### 2.1 Integrated cognitive assessment (ICA) task

ICA [44, 50], [51, p.] is a rapid animal vs. non-animal categorization task. The participants are presented with natural images of animals and non-animals. The images are followed by a short blank screen and then a dynamic mask. Participants should categorize the images as animal or non-animal as quickly and accurately as possible (Fig 1). See (Khaligh-Razavi et al. 2019 [44], Fig 1B) for sample images of the task.

### 2.2 Montreal cognitive assessment (MoCA)

MoCA [52] is a ten-minute pen and paper test with a maximum score of 30 and is conventionally used to assess visuospatial, memory, attention, and language abilities to detect cognitive impairment in older adults. An examiner is needed to administer the test. The results of this test were used by the consultant neurologist to help with the diagnosis of participants.

### 2.3 Addenbrooke cognitive examination (ACE-R)

ACE-R [53] is another pen-and-paper tool for cognitive assessment with a maximum score of 100. It mainly assesses five cognitive domains: attention, orientation, memory, fluency, language, and visuospatial. On average, the test takes about 20 to 30 minutes to administer and score. The results of this test were used by the consultant neurologist to help with the diagnosis of participants.

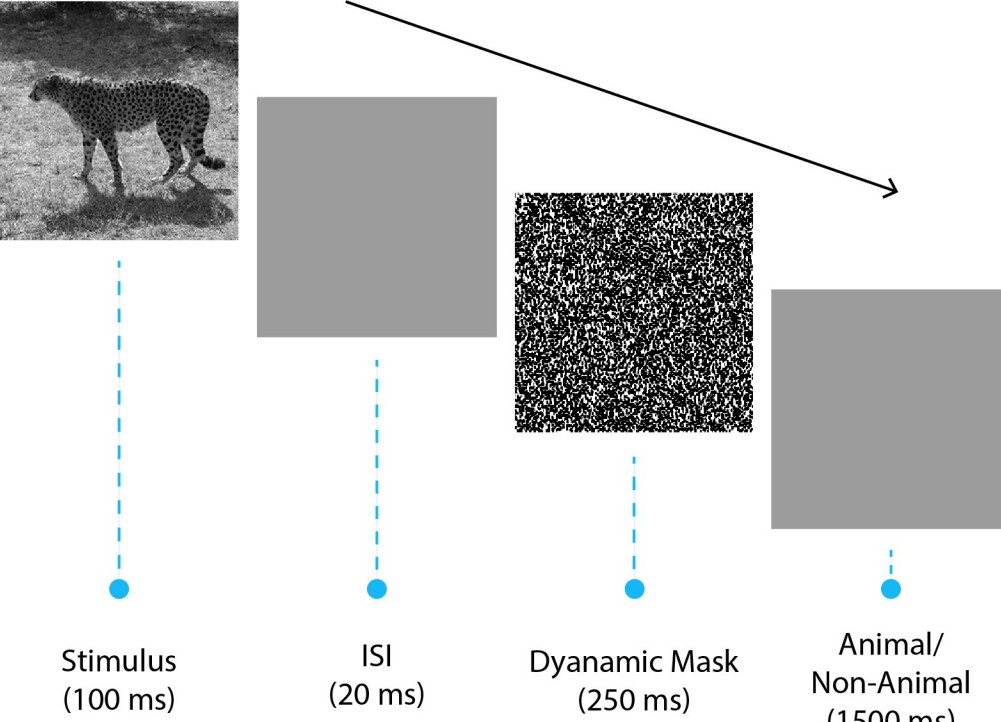

Stimulus
(100 ms)

ISI
(20 ms)

Dyanamic Mask
(250 ms)

Animal/
Non-Animal
(1500 ms)

**Fig 1. The EEG task.** The EEG experiment contained 13 experimental runs; in each run, 32 natural images (16 animal, 16 non-animal) were presented to the participant in a random order. Each image was shown for 100 ms, followed by an inter-stimulus interval (ISI) of 20ms and a dynamic noise mask for 250ms. Participants were given 1500 ms to indicate (using pre-specified buttons) whether the presented image contained an animal or not.

## 2.4 Subject recruitment

40 (22 Healthy, 18 MCI) participants (Table 1) completed the ICA test, MoCA, and ACE-R in the first assessment. The participants were non-English speakers, with instructions for the cognitive assessments provided in Farsi. The ICA test was administered on an iPad.

Patients were recruited into the study prospectively. A consultant neurologist made all the diagnoses according to diagnostic criteria described by the working group formed by the National Institute of Neurological and Communicative Disorders and Stroke (NINCDS) and the Alzheimer's Disease and Related Disorders Association (ADRDA) (referred to as the NINCDS-ADRDA criteria) and the National Institute on Aging and Alzheimer's Association (NIA-AA) diagnostic guidelines. The study was conducted at the Royan Institute, according to the Declaration of Helsinki, and approved by the local ethics committee at the Institute. The inclusion/exclusion criteria are listed below.

- Inclusion criteria for the HC group:
  Males and females aged between 50–85 years who are not currently on medication that may interfere with the study results and are in good general health were included in the study.

- Inclusion criteria for the MCI group:
  Males and females aged between 50–85 years with a clinical diagnosis of MCI who were able to provide informed consent were included in the study.

- Exclusion criteria for both groups:
  Individuals with a presence of significant cerebrovascular disease or major psychiatric disorder (e.g., chronic psychosis, recurrent depressive disorder, generalized anxiety disorder, and bipolar mood disorder) or major medical comorbidities (e.g., congestive cardiac failure, diabetes mellitus with renal impairment) were excluded from the study.
  Additional exclusion criteria were: use of cognitive-enhancing drugs (e.g., cholinesterase inhibitors), or a concurrent diagnosis of epilepsy or any history of alcohol misuse, illicit drug abuse, severe visual impairment (e.g., macular degeneration, diabetic retinopathy, as determined by the clinical examination), or repeated head trauma.

## 2.5 EEG data acquisition and preprocessing

The EEG experiment included a short version of the ICA task (i.e., smaller image set). Participants completed one EEG session that included 13 runs; each run lasted 67 seconds, during which 32 images were presented in random order, and participants had a short break in between the runs. Each stimulus was repeated 13 times during the whole EEG session (once in each run). These are referred to as repetition trials throughout the manuscript. Participants 16 and 17 completed 10 runs, and participants 12 and 22 completed 12 runs. We acquired the EEG data using a 64-channel (63 electrodes on the cap and one as the reference; for electrodes layout, see S1 Fig of the supplementary materials) g.tec product at a sampling rate of 1200 Hz. Three electrooculograms (EOG) channels were set up to capture eye blinks. Additionally, we

**Table 1. Demographic information of participants.**

| Characteristic | HC (*n* = 22) | MCI (*n* = 18) | *P-values* |
|---|---|---|---|
| Age–mean years ±SD | 63.41±6.10 | 63.56±6.41 | 0.94 |
| Education in years–mean ±SD | 15±4.18 | 14.72±5.02 | 0.85 |
| Gender (%female) | 13 (59%) | 10 (55%) | 0.82 |

SD: standard deviation.

p-values were calculated from two-sided t-test for two independent samples.

included resting trials in between the image trials (i.e., almost every 70 seconds, they were given 10 seconds to rest their eyes, blink, and swallow). Participants were instructed to only blink (or swallow) during these trials to prevent contamination of EEG signals with the eye-blink (and swallowing) artifacts. These trials were excluded from further EEG analyses. Because of such a design, we did not have to reject any of the image trials. Other potential artifacts were removed with Independent Component Analysis.

The preprocessing consisted of six general steps, which were all done using Brainstorm [54] in MATLAB:

1. Re-referencing the data with the mean activation and removing the reference channel (channel 33).

2. Neutralizing eye blinks by removing the most correlated component with the EOG channels, using the independent component analysis algorithm.

3. Extracting pre-stimulus data from 100 ms before to 800 ms after the stimulus onset (epoching).

4. Normalizing the epochs regarding the mean and standard deviation of the baseline.

5. Smoothing the data with a 50 Hz low-pass filter.

6. Resampling the data to 1000 Hz sampling rate.

## 2.6 Univariate pattern analysis—Event-Related Potential (ERP)

We calculated the ERPs of the extracted epochs (from 100 ms before to 800 ms after the stimulus onset) by averaging the EEG responses to all stimuli within each group of channels. We calculated the ERPs separately for each individual. The ERP of HC and MCI are the average ERP of corresponding participants.

## 2.7 Support vector classifier

We used a linear support vector machine (SVM) classifier throughout the analyses to decode neural data. The hyperparameters were as follows: The cost/regularization parameter (C) and the weight of classes were all set to 1. All the classifications were done using the LIBSVM software implementation [55].

## 2.8 Multivariate pattern analysis—Animal vs. non-animal decoding

To study the emergence of animacy categorical information in the brain, we used a linear classifier to discriminate EEG responses to animal stimuli from that of non-animal (Fig 2). Before the classification, we randomly assigned each target stimulus with all its EEG trials to bins of sizes 2, 3, and 4 stimuli and randomly sub-averaged the trials within each bin. The decoding accuracy at each time point 't' is then the average accuracy of 10,000 repetitions in a leave-one-bin-out cross-validation model, using an SVM classifier.

## 2.9 Multivariate pattern analysis—Pairwise decoding

At each time point, we measured the accuracy of an SVM classifier in discriminating pairs of stimuli using leave-one-out cross-validation. This leads to a symmetric 32×32 representational dissimilarity matrix (RDM) at each time point, representing the pairwise dissimilarities of stimuli in the off-diagonal elements (Fig 3). We repeated the entire process for each individual to create an RDM at each time point.

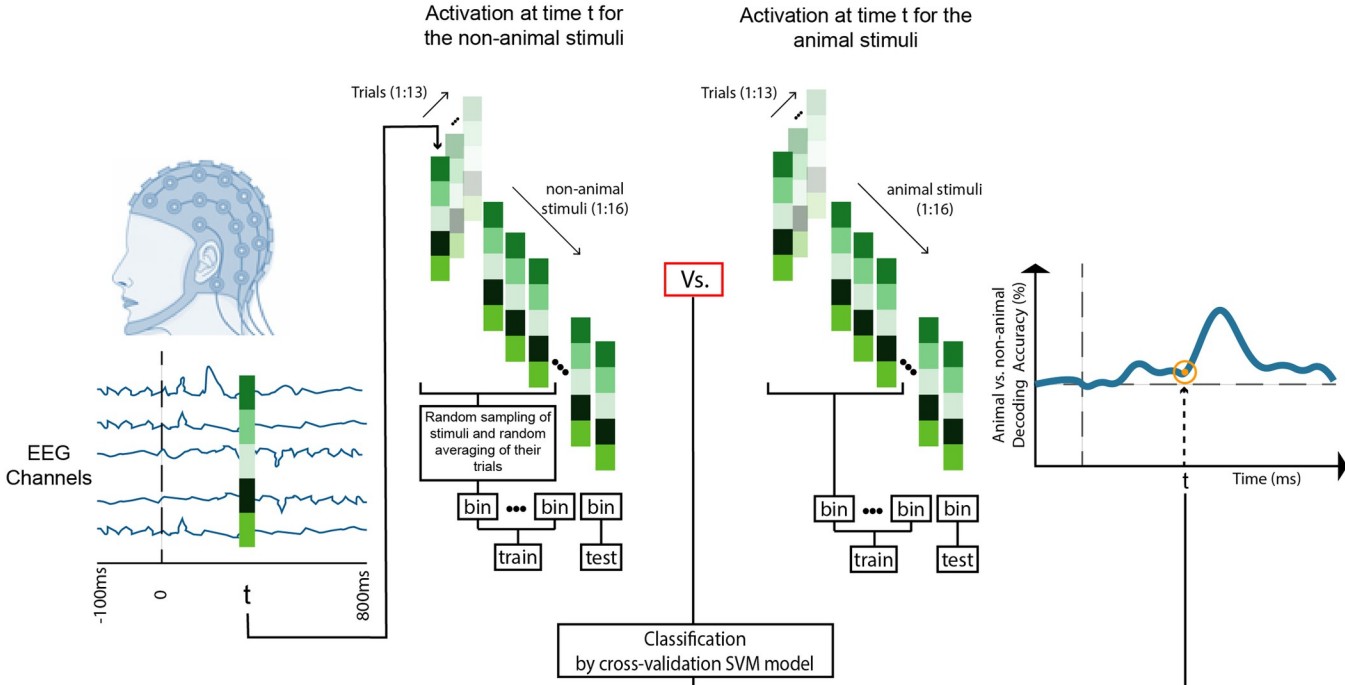

**Fig 2. Animal vs. non-animal decoding.** We extracted repetition trials of EEG responses to animal and non-animal stimuli at each time-point (for a given time-point t we had two matrices of 63 channels x16 stimuli x13 repetition trials of EEG responses to animal and non-animal stimuli). Before the classification, we randomly assigned each target stimulus with all its EEG trials to bins of size 2, 3 and 4 stimuli and randomly sub-averaged the trials within each bin. We trained a leave-one-bin-out cross-validation SVM model to discriminate animal from non-animal. At each time-point, the decoding accuracy is the average of 10,000 repetitions of the classification procedure described above. We repeated the entire process of each individual separately.

## 2.10 Multivariate pattern analysis—HC vs. MCI classification

We characterized the activation pattern at each time-point as a 63×32 matrix, with each column being the responses of the EEG channels to a stimulus, averaged over all repetition trials (Fig 4). We applied 10,000 bootstrap resampling (without replacement) of participants and trained a leave-one-out cross-validation SVM model to discriminate the EEG activation pattern of HC from that of MCI.

## 2.11 Multidimensional Scaling (MDS)

Multidimensional Scaling (MDS) is a non-linear dimension reduction algorithm. It rearranges the data points in a p-dimensional space until their pairwise distance is consistent with a given dissimilarity matrix. Here, we used MDS to visualize the stimuli on a 2D plane based on their pairwise dissimilarity in RDMs.

## 2.12 Statistical analysis

To avoid any assumption about the observed distributions, we only used non-parametric statistical tests. They are capable of testing a null hypothesis without any prior assumptions about the nature of the distribution:

**2.12.1 Bootstrap test.** Bootstrapping is a strategy to estimate different statistics over an unknown distribution. It consists of a resampling (with or without replacement) procedure and applying a target function. The result is a bootstrap distribution that can be used for statistical inference purposes. We used 10,000 bootstrap resampling of participants without

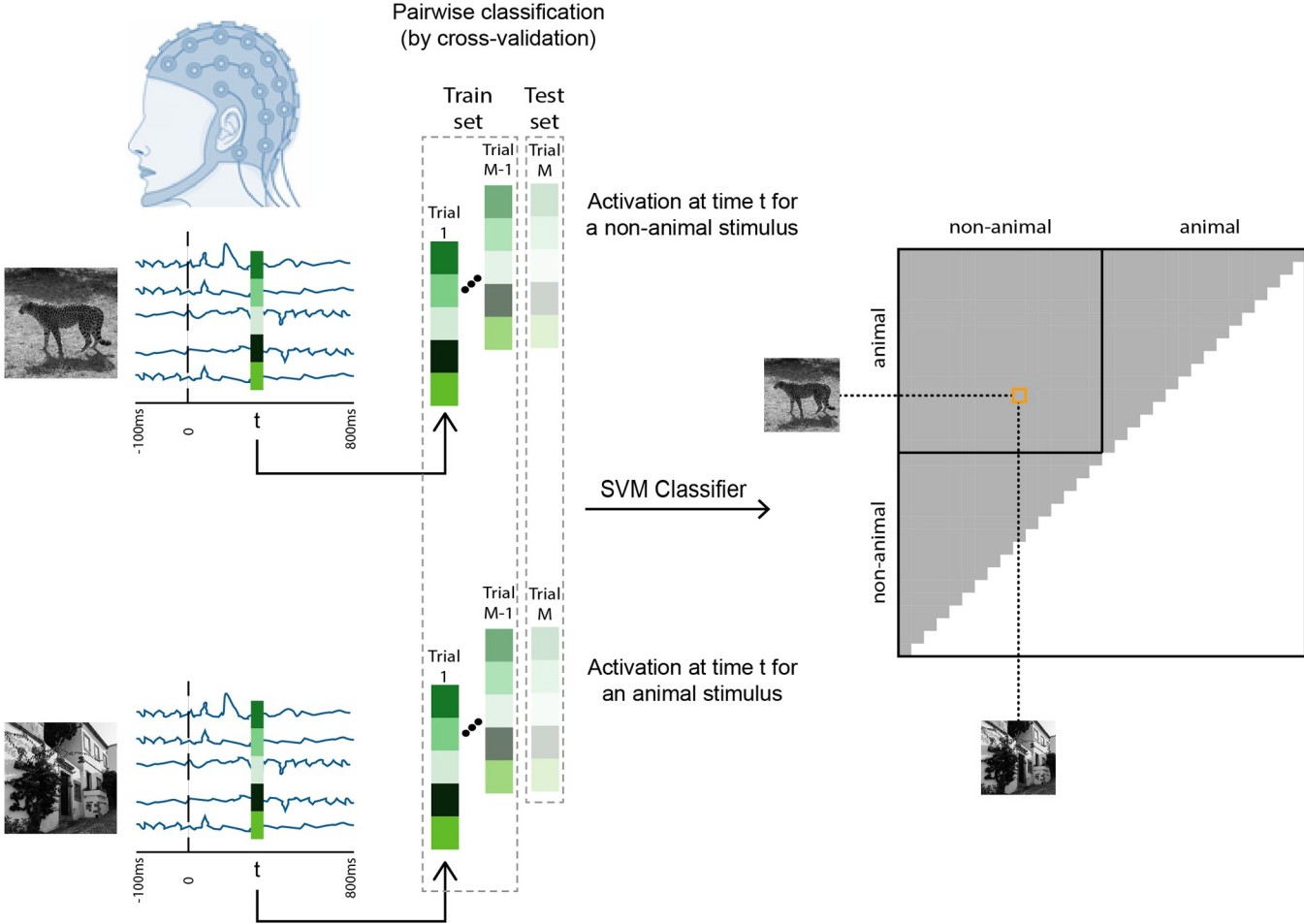

**Fig 3. Representational dissimilarity matrices (RDMs).** We trained an SVM to discriminate pairs of stimuli using their EEG responses at time-point t. This pairwise classification of stimuli by cross-validation SVM model leads to a 32 × 32 RDM at each time-point t.

replacement from each group (HC and MCI) and computed a p-value as follows:

$$p-value = \frac{number\ of\ elements\ lower\ (or\ higher)\ than\ baseline + 1}{number\ of\ bootstrap\ resampling\ (10000) + 1}$$

**2.12.2 Permutation test.** The permutation test consists of randomly relabeling the samples from two populations to form a null distribution. It computes a p-value by testing a target statistic against the null hypothesis:

$$p-value = \frac{number\ of\ members\ from\ the\ null\ distibution\ lower\ (or\ higher)\ than\ the\ target + 1}{number\ of\ permutations\ (10000) + 1}$$

**2.12.3 Rank-sum.** Rank-sum (also known as Wilcoxon–Mann–Whitney test) tests the null hypothesis that the data in $x$ and $y$ are sampled from continuous distributions with equal medians, against the alternative that they are not [56]. We used rank-sum to compute the p-value when comparing HC and MCI medians of animal vs. non-animal decoding amplitude, ICA speed and accuracy and, mean of ERP responses.

HC vs. MCI Classification
(by cross-validation)

**Fig 4. HC vs. MCI classification over time based on EEG response patterns.** At each time-point, the pattern of EEG activation is a 63x32 matrix with columns being the EEG responses of channels to the 32 stimuli. We applied 10,000 bootstrap resamplings (without replacement) of participants and each time trained a leave-one-out cross-validation SVM model to discriminate HC from MCI based on their EEG activation patterns.

## 3 Results

### 3.1 A reduction in the speed and accuracy of animacy processing in MCI patients

We compared the neural speed and accuracy of animal/non-animal discrimination between HC and MCI. To this end, for each group, we computed the time at which animal images can best be discriminated from non-animals based on their EEG responses. This time-point is referred to as the peak of animal/non-animal decoding. MCI patients showed a median delay of 39 ms (95% CI = [8, 111], SE = 34 ms) in processing the animacy information in comparison to healthy individuals (p-value = 0.0001; 10000 bootstrap resampling of participants, Fig 5A). Additionally, in this decoding peak time-point, the neural accuracy of animal detection was

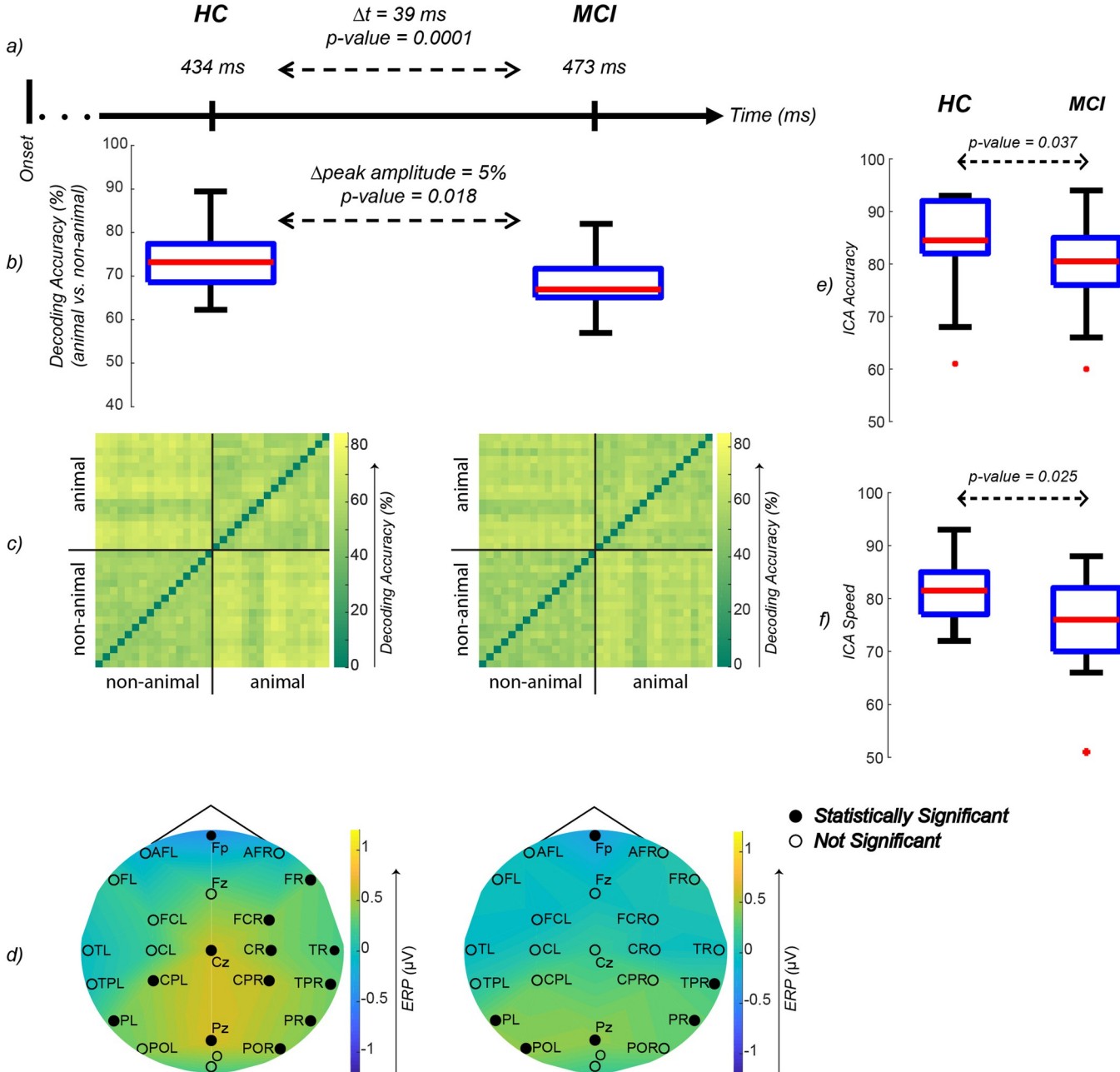

**Fig 5. a)** Median of animal vs. non-animal decoding accuracy peak time-points. 10000 bootstrap resampling without replacement of participants was applied to measure the differences between HC and MCI. The difference of medians of HC and MCI peaks is 39 ms (p-value = 0.0001; One-sided bootstrap test) **b)** Box-plot of animal vs. non-animal EEG decoding peak amplitude. The difference of HC and MCI amplitude medians is 5% (p-value = 0.018; rank-sum). **c)** Mean EEG RDM of participants at the time of animal vs. non-animal decoding peak. No significant difference was observed between the RDMs (permutation test). **d)** Mean ERP of participants at the time of animal vs. non-animal decoding peak. The black dots indicate significant activation in channels of the specified region (FDR-corrected at 0.05 sign-rank). No significant difference between the ERP of HC and MCI was observed (FDR-Corrected at 0.05 rank-sum). **e)** Box-plot of the ICA test accuracy (results of the behavioral ICA, taken on iPad). A significant difference is observed between HC and MCI (p-value = 0.037; rank-sum). **f)** Box-plot of the ICA test speed (results of the behavioral ICA, taken on iPad). A significant difference is observed between HC and MCI (p-value = 0.025; rank-sum). Fp: prefrontal; AFL: left anterior frontal; AFR: right anterior frontal; FL left frontal; Fz: midline frontal; FR: right frontal; FCL: left fronto-central; FCR: right fronto-central; TL: left temporal; CL: left central; Cz: midline central; CR: right central; TR: right temporal; TPL: left temporo-parietal; CPL: left centro-parietal; CPR: right centro-parietal; TPR: right temporo-parietal; PL: left parietal; PR: right parietal; POL: left parieto-occipital; Pz: midline parietal; POR: right parieto-occipital; O: occipital.

significantly lower in MCI patients compared to healthy controls (p-value = 0.018; rank-sum, Fig 5B).

The representational dissimilarity method is a way to represent patterns of brain information processing [57]. To study the neural representation underlying animacy categorization, we compared the representational dissimilarity matrices (RDM) and ERP responses (Fig 5C and 5D) of the two groups at the time of each individual's animal vs. non-animal decoding peak.

The HC/MCI RDM in the peak animacy decoding time-point represents the pattern of EEG responses at a time-point in which the brain representation of animal images is best separated from that of non-animals (Fig 5C). While the peak animacy time-point was delayed for the MCI group, the MCI RDM was not significantly different from that of the HC RDM in its peak, suggesting that it took more time for the MCI patients to converge to a pattern similar to that of the HC, which is used for discriminating animals from non-animals.

We also looked at the ERP responses at these peak time-points between HC and MCI: channels in the right and left parietals, right temporal-parietal, midline parietal, and midline frontal lobes were significantly activated in both groups. HC showed significant activation levels in the right parieto-occipital, right temporal, and central lobes (midline central, right and left central-parietals, right central, and fronto-central); however, these were absent from the MCI brain activation map. On the other hand, only MCI patients showed significant activation on the left parieto-occipital lobe.

Analyzing the behavioral data of participants while taking the ICA test (on an iPad outside the EEG), we found the results to be consistent with the findings from the EEG data: the speed and accuracy of animacy detection, as measured by the ICA test, were also significantly deteriorated in patients with MCI (p-value = 0.025 and p-value = 0.037 respectively) (Fig 5F and 5G).

We also examined the channel-specific animacy decoding time courses to investigate whether there is a significant delay in the peak and/or the onset of animacy processing in MCI patients at the level of EEG channels (Fig 6). We found that there is a significant delay in the animacy decoding peak time-point of MCI patients in the right parietal lobe compared to HC (p-value = 0.0013, Fig 6). Additionally, the significance of animal vs. non-animal decoding started later in the MCI patients in EEG channels of left fronto-central, midline frontal, midline central, and left parieto-occipital lobes (Fig 6).

## 3.2 Comparing patterns of visual information processing in HC and MCI

We compared patterns of visual information processing in HC and MCI using their RDMs over time. The maximum difference between HC and MCI RDMs was observed at t = 224 ms after the stimulus onset (scaled Euclidean distance, d = 0.53 [out of 1], p-value = 0.012; Fig 7A). At this time-point (t = 224 ms), an SVM classifier (leave-one-out cross-validation) could significantly discriminate between HC and MCI patterns of visual information (represented by their RDMs) with an accuracy of 70.4% (p-value = 0.0036, 10,000 bootstrap sampling of participants).

We also looked at the univariate differences between HC and MCI at the same time-point (i.e., t = 224 ms); there was no significant difference between the two groups in their ERP responses (Fig 7B). This demonstrates that the response patterns carry valuable information above and beyond what is captured in the neural activity's mean responses.

Furthermore, examining the HC and MCI RDMs, we highlighted RDM cells that were significantly different between the two groups (Fig 7A). Each RDM cell represents the difference between the EEG patterns of the two stimuli at a given time-point. To provide a more intuitive

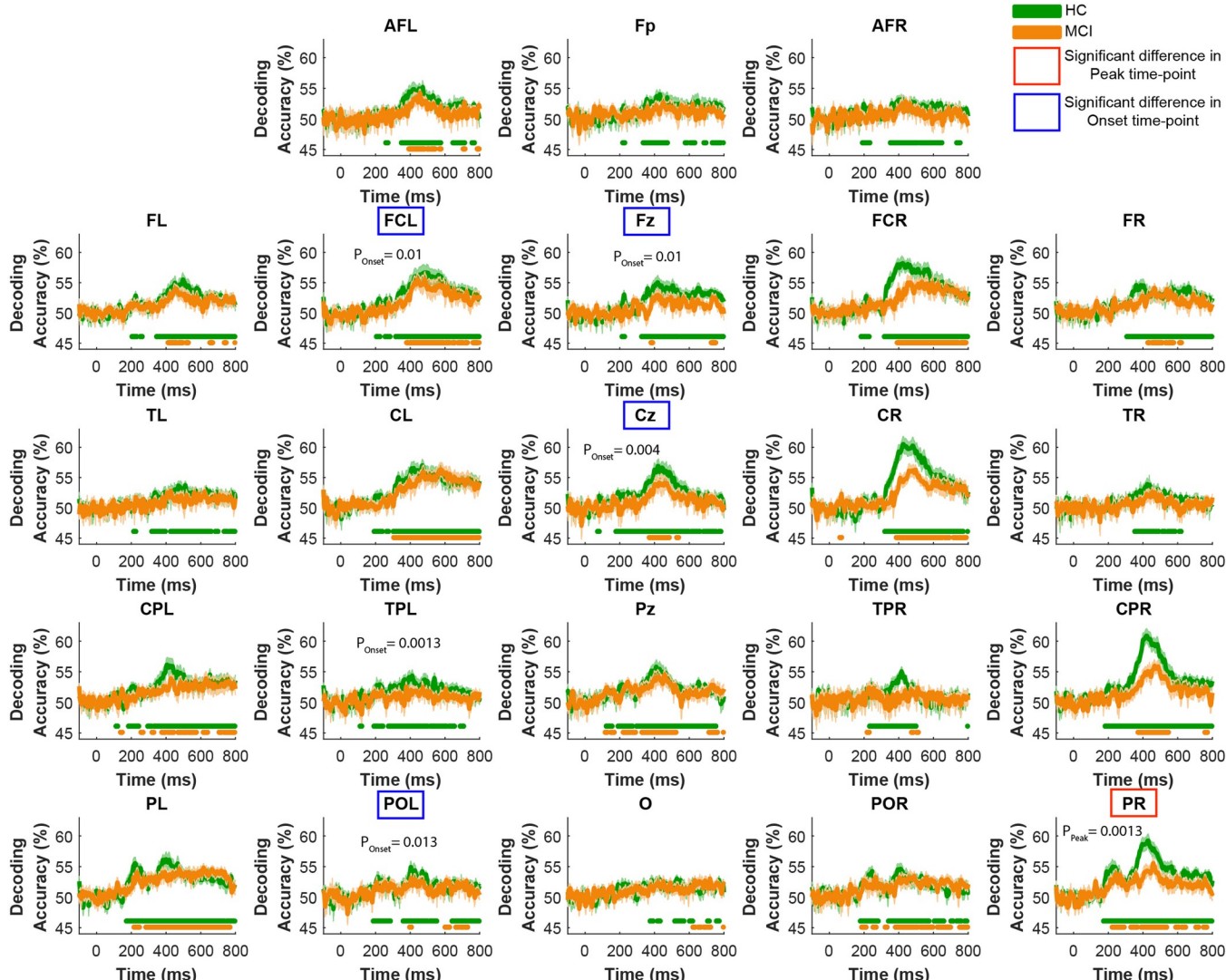

**Fig 6. Animal vs. non-animal decoding time course across groups of EEG channels.** Horizontal color dots indicate time-points with significant decoding accuracy in the corresponding group (green for HC and orange for MCI). In regions specified with red rectangles MCI's peak animacy decoding time-point is significantly later than that of HC; in regions specified with blue rectangles MCI's onset of animacy decoding significance is significantly later than that of HC (FDR-corrected at 0.05; bootstrap test with 10,000 resampling of participants on MCI minus HC time-points).

understating of the differences in patterns, we used multidimensional scaling (MDS) to visualize the RDMs on a 2D surface. Fig 7C illustrates the stimuli with a significant difference between HC and MCI, and Fig 7D visualizes all the stimuli.

### 3.3 Temporal dynamics of animacy categorization

To study the temporal dynamics of animacy categorization in HC and MCI, we compared the mean response (i.e., ERP) as well as EEG activation patterns (i.e., 63 channels × 32 stimuli matrices) between HC and MCI over time. For each EEG channel group, we measured the ERP differences between HC and MCI over time: midline frontal, right fronto-central, and right temporal regions showed a significant difference (Fig 8A). We also performed a classification over the EEG activation patterns to see if HC and MCI can be discriminated based on

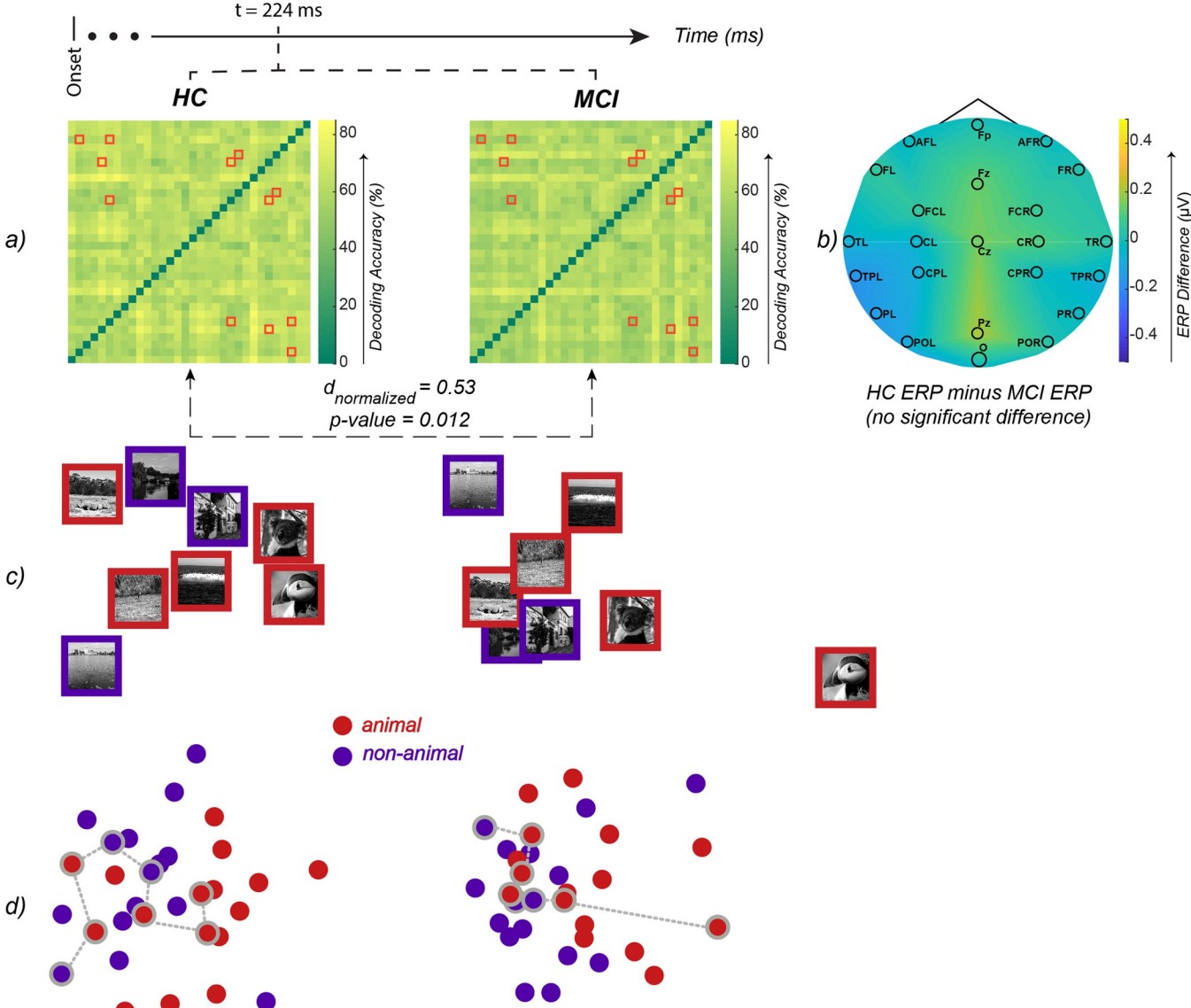

**Fig 7.** a) The RDM of HC and MCI at the time-point of maximum Euclidean distance (t = 224 ms, d = 146.6, p-value = 0.012, permutation test). We further normalized the Euclidean distance to make it more meaningful by fitting a logarithmic function on the maximum distance (222.7), baseline distance (115.7) and the minimum distance (0). The logarithmic function scales the distances to the [0, 1] interval, and the observed distance between RDMs becomes 0.53. The highlighted elements (i.e. red-squares) of the RDMs are the pairwise dissimilarities with a significant difference between HC and MCI (FDR-corrected at 0.05 rank-sum). b) Difference of the mean ERPs across EEG channels (HC minus MCI) at t = 224 ms. None of the EEG channels show a significant difference (all p-values > 0.05) between HC and MCI at this time-point. c) MDS of the stimuli that showed a significant difference in their pairwise dissimilarity between HC and MCI (those specified by red squares in the two RDMs). d) MDS of all the stimuli. The dots that are connected with dashed lines are the same stimuli shown in panel 'c'.

their epoched EEG responses. HC and MCI could be discriminated based on their EEG activation patterns in the left frontal, midline frontal, left parietal, and right central parietal lobes (Fig 8B).

Looking at the EEG data (Fig 8), we found that HC and MCI could be discriminated starting from 375 ms in the left parietal (PL) and from 495 ms in the left frontal (FL) both to 515 ms after the stimulus onset only based on their patterns of activity, but not the ERPs. Additionally, the pattern of activity in centro-parietal (CPR) could discriminate HC from MCI in

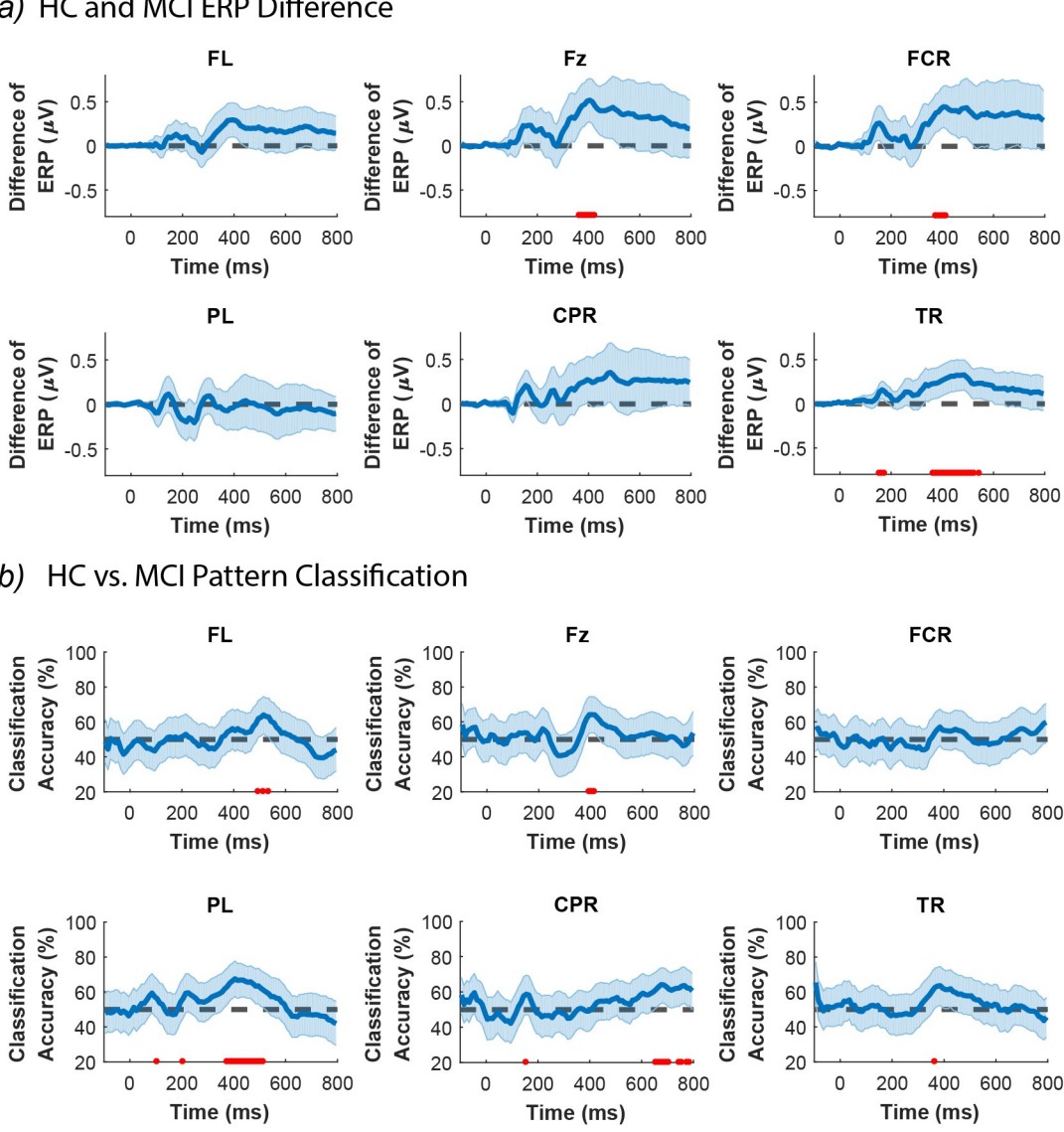

**Fig 8.** a) HC and. MCI ERP differences in regions where either the ERP difference or the HC vs. MCI classification were statistically significant. The shaded error bars indicate 95% confidence interval of the ERP difference (HC minus MCI). Red dots indicate time-points with significant level of difference in ERP (FDR-corrected at 0.05 across time; rank-sum). b) The HC vs. MCI classification based on the pattern of EEG data (i.e. channels × stimuli), in regions where either the ERP difference or the HC vs. MCI classification of EEG responses were statistically significant. The shaded error bars indicate 95% confidence intervals. Red dots indicate time-points with significant HC vs. MCI classification accuracy (FDR-corrected at 0.05; 10000 bootstrap resampling of participants). FL: left frontal; Fz: midline frontal; FCR: right fronto-central; PL: left parietal; CPR: right central parietal; TR: right temporal.

almost every time-point after t = 655ms. On the other hand, the ERP responses showed a significant difference between HC and MCI in the right fronto-central (FCR) at around t = 405 ms and in the right temporal (TR) from 155 ms to 174 ms and 365 ms to 545 ms. At the same time points, the two groups could not be separated based on their activation patterns. The midline frontal (Fz) was the only region that could differentiate HC and MCI based on both the ERP responses and the activation patterns at around t = 405 ms after the stimulus onset.

In the previous subsection, we demonstrated that at t = 224 ms, the difference between HC and MCI in the EEG response patterns (captured by RDM) was at its maximum, while the level of activity (captured by ERP) remains unchanged. Here, we identified five groups of channels whose EEG data could discriminate between HC and MCI, either based on the activation patterns or the ERP responses, but not both. This is consistent with the reported results in the previous section (3.2), demonstrating that EEG activation patterns could be different even though there might be no difference at the level of ERP.

## 4 Discussion

In this study, we investigated the temporal dynamics of animacy visual processing in patients with MCI and argued that the speed of processing animacy information is a potential bio-marker for detecting MCI. The proposed rapid visual categorization task in the ICA test is more challenging than the typical memory tasks, making it more sensitive to less severe brain deteriorations. The ICA is particularly suited for population-wide screening of cognitive impairment to help identify patients with MCI and mild AD (MiAD) and not designed for cognitive assessment in more severe stages of the disease, such as moderate to severe AD. Early detection of cognitive impairment is becoming increasingly important, particularly in the context of disease-modifying therapies for early stages of AD, such as Aducanumab–which has recently received FDA approval.

Previous resting-state and task-based EEG studies have focused on univariate changes (e.g., ERP, frequency bands, connectivity) in patients with MCI and individuals with mild to moderate AD [5–7, 15–17, 58]. Here, using a rapid visual categorization task and applying multivariate pattern analysis, we looked beyond the univariate changes and studied the categorical representation of animacy information in the brain of old healthy individuals and patients with MCI. We demonstrated that patients with MCI could be discriminated from HC based on their pattern of animacy representation. Furthermore, we identified regions in which either the mean EEG responses or the pattern of brain activity show significant differences between HC and MCI.

Having a closer look at the ERP of different regions, we observed a decrease in the P300 amplitude of the MCI patients. This decrease is significant in the electrodes of the midline frontal, right fronto-central, and right temporal regions between 250 ms to 500 ms after the stimulus onset. This observation is in line with the reported result in [13], especially with the P300 of the midline frontal electrodes, and highlights the importance of this signal in the detection of MCI.

### 4.1 Task differences in EEG studies of HC and MCI

Consistent with previous reports in resting-state EEG studies [59] and studies with a visual memory task [17], we observed univariate differences between HC and MCI in the temporal and the fronto-central electrodes.

In contrast, we did not find any significant difference in ERP responses of HC vs. MCI in centro-parietal and parietal electrodes–which is reported previously in an EEG study with a visual working memory task [16]. We also observed no univariate difference in the frontal and occipital electrodes, while previous resting-state EEG studies have reported differences between HC and MCI in these regions [59, 60]. On the other hand, we found that MCI patients could be discriminated from HC based on their ERP responses of the midline frontal region electrodes. These differences could potentially be explained by the difference in the tasks used for each of these studies (i.e., visual working memory and resting-state vs. rapid visual categorization).

In addition to the previous univariate findings, here we revealed multivariate differences between HC and MCI in their patterns of EEG responses: midline frontal, left frontal, left parietal, and right centro-parietal electrodes showed significant multivariate differences between HC and MCI. Furthermore, the categorical representation of animacy information of the right parietal electrode emerged later in the MCI patients compared to that of HC. Also, in comparison with HC, the MCI patients had significantly longer onset latencies of animacy information in the left fronto-central, midline frontal, midline central, and left parieto-occipital electrodes.

## 4.2 What do differences in the pattern of activation mean?

The overall changes in the pattern of EEG responses happen in the regions that are involved in visual processing and motor movement. These regions are engaged during the ICA task's execution, and their activation is captured through frontal and parietal electrodes.

Neurons of the parietal cortex integrate sensory inputs (visual, auditory, etc.) through motor control regions to execute movements [61–63]. Visuomotor skills that are known to be resolved in regions of the parietal cortex are heavily involved in the ICA task, as the task requires the participant to use visual information to categorize presented images with a movement of both hands.

In the case of frontal regions, univariate changes in the electrodes of this lobe have previously been shown in EEG studies [17, 59]. Additionally, the Posterior-Anterior Shift in Aging (PASA) suggests that the brain of individuals with age-related changes tends to engage other networks in the frontal region to compensate for the decline of processing information in posterior areas [64]. Consistent with PASA, we observed that the pattern of information processing is altered in both frontal and parietal electrodes.

## 4.3 Neural speed of information processing in MCI patients

Rapid recognition of animate objects is a fundamental ability of human visual cognition. Previous M/EEG studies have investigated the temporal neural dynamics of animacy processing in young, healthy individuals. Using slightly different visual tasks and stimuli, studies have shown that the onset and peak of animacy decoding emerge between 66 ms to 157 ms [35]; or from 80 ms to 240 ms [36] after the stimulus onset. Here, we showed that in old healthy individuals, the onset, and the peak of animacy decoding emerge between 131 ms (SE = 30) and 434 ms (SE = 30) after the stimulus onset. This result indicates that normal aging causes a reduction in the animacy information processing speed (IPS). Compared to the old healthy individuals, animacy IPS was further delayed in MCI patients (onset of animacy decoding: 196 ms±16, peak animacy decoding: 473 ms±34 after the stimulus onset). These findings confirm a significant decrease in the speed of neural information processing in patients with MCI and are consistent with previous behavioral studies showing a decline in visual IPS in MCI patients compared to HC [44, 47]. Together, these results suggest the IPS as a potential biomarker for the detection of MCI patients. However, other complementary biomarkers should be employed for the diagnosis of MCI due to AD.

Some of the study limitations include the relatively small number of patients recruited. Additionally, the study was not a longitudinal study to determine if the MCI patients will convert to AD. In future longitudinal studies, we aim to investigate how well the current results generalize to cohorts of larger MCI/AD patients.

## 5 Conclusion

We showed that the speed of processing animacy information is decreased in MCI patients compared to healthy individuals. This suggests the use of the ICA test, which is based on the

categorization of animal and non-animal images, as a digital biomarker for detecting cognitive impairment in MCI patients.

Furthermore, we showed that in addition to univariate changes, the brains of MCI patients and HC individuals are different in the pattern of representing animacy information.

## Supporting information

**S1 Fig. EEG electrodes layout.** We used a 64-channel g.tec product at a sampling rate of 1200 Hz for EEG data acquisition (S1 Fig). The reference electrode (#33) was placed on the participant's right ear. Fp: prefrontal; AFL: left anterior frontal; AFR: right anterior frontal; FL left frontal; Fz: midline frontal; FR: right frontal; FCL: left fronto-central; FCR: right fronto-central; TL: left temporal; CL: left central; Cz: midline central; CR: right central; TR: right temporal; TPL: left temporo-parietal; CPL: left centro-parietal; CPR: right centro-parietal; TPR: right temporo-parietal; PL: left parietal; PR: right parietal; POL: left parieto-occipital; Pz: midline parietal; POR: right parieto-occipital; O: occipital.
(TIF)

**S1 Table. Performance of participants on paper tests (MoCA and ACE-R) and ICA test (administered on iPad).** The absent subjects were removed from this study either due to their status of disease (AD or mild AD) or that their status changed during the development of the study.
(PDF)

## Acknowledgments

We thank the National Brain Mapping Laboratory (NBML), where all the EEG data acquisitions were done. Authors are also grateful to Mohammad Mohaghar for proofreading the manuscript.

## Author Contributions

**Conceptualization:** Seyed-Mahdi Khaligh-Razavi.

**Data curation:** Hamed Karimi, Haniyeh Marefat, Mahdiyeh Khanbagi, Chris Kalafatis, Zahra Vahabi.

**Formal analysis:** Hamed Karimi.

**Funding acquisition:** Seyed-Mahdi Khaligh-Razavi.

**Investigation:** Hamed Karimi.

**Methodology:** Seyed-Mahdi Khaligh-Razavi.

**Project administration:** Seyed-Mahdi Khaligh-Razavi.

**Supervision:** Seyed-Mahdi Khaligh-Razavi.

**Writing – original draft:** Hamed Karimi.

**Writing – review & editing:** Haniyeh Marefat, Mahdiyeh Khanbagi, Chris Kalafatis, Mohammad Hadi Modarres, Zahra Vahabi, Seyed-Mahdi Khaligh-Razavi.

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
