## [Decision Letter · Decision Letter 0]

6 May 2021

PONE-D-21-04261

Temporal dynamics of animacy categorization in the brain of patients with mild cognitive impairment

PLOS ONE

Dear Dr. Karimi,

Thank you for submitting your manuscript to PLOS ONE. After careful consideration, we feel that it has merit but does not fully meet PLOS ONE’s publication criteria as it currently stands. Therefore, we invite you to submit a revised version of the manuscript that addresses the points raised during the review process.

I have received the comments from two reviewers. One of the two reviewers raised an issue with the small sample size of patients group.

In my opinion, you can estimate the power of your analysis and also you can add Cohen's d statistic while

you can avoid strong use of significance.

A paragraph with limitation should be added in the discussion part.

If you decided to revise according to reviewers' comments, you should update the manuscript and prepare

a response letter to every comment.

We look forward to receiving your revised manuscript.

Kind regards,

Stavros I. Dimitriadis

Academic Editor

PLOS ONE

Additional Editor Comments:

Dear authors,

I have received the comments from two reviewers. One of the two reviewers raised an issue with the small sample

size of patients group.

In my opinion, you can estimate the power of your analysis and also you can add Cohen's d statistic while

you can avoid strong use of significance.

A paragraph with limitation should be added in the discussion part.

If you decided to revise according to reviewers' comments, you should revise the manuscript and prepare

a response letter to every comment.

Journal Requirements:

"SKR and CK serve as the CSO and CMO at Cognetivity ltd., and MM as the lead data scientist. Other authors declared no competing interests."

We note that one or more of the authors are employed by a commercial company: Cognetivity ltd.

2.1. Please provide an amended Funding Statement declaring this commercial affiliation, as well as a statement regarding the Role of Funders in your study. If the funding organization did not play a role in the study design, data collection and analysis, decision to publish, or preparation of the manuscript and only provided financial support in the form of authors' salaries and/or research materials, please review your statements relating to the author contributions, and ensure you have specifically and accurately indicated the role(s) that these authors had in your study. You can update author roles in the Author Contributions section of the online submission form.

2.2. Please also provide an updated Competing Interests Statement declaring this commercial affiliation along with any other relevant declarations relating to employment, consultancy, patents, products in development, or marketed products, etc.  

Reviewers' comments:

Reviewer's Responses to Questions

**Comments to the Author**

1. Is the manuscript technically sound, and do the data support the conclusions?

Reviewer #1: Yes

Reviewer #2: Partly

2. Has the statistical analysis been performed appropriately and rigorously? 

Reviewer #1: Yes

Reviewer #2: I Don't Know

3. Have the authors made all data underlying the findings in their manuscript fully available?

Reviewer #1: Yes

Reviewer #2: No

4. Is the manuscript presented in an intelligible fashion and written in standard English?

Reviewer #1: Yes

Reviewer #2: Yes

5. Review Comments to the Author

Reviewer #1: Overall:

The manuscript analyzes the ERP response during the integrated cognitive assessment to find differences between healthy controls and patients with mild cognitive impairment.

Results suggest a slower response in the MCI population.

Recommend:

Major comments:

* In Section 1, relevant works on the use of EEG for AD and MCI assessment are missing, e.g.:

- Regarding resting state EEG: "Systematic Review on Resting-State EEG for Alzheimer’s Disease Diagnosis and Progression Assessment Resting-state EEG" (Cassani, 2018), and

- Regarding ERPs: "P300 Amplitude in Alzheimer’s Disease: A Meta-Analysis and Meta-Regression" (Hedges, 2016)

- Regarding slower EEG signals: "Slowing and Loss of Complexity in Alzheimer's EEG: Two Sides of the Same Coin?" (Dauwels, 2011)

* The MoCA and ACE-R tests are described in Sections 2.2 and 2.3 respectively. In Section 2.4, it is stated the participants completed these tests, however the scores are not presented nor used.

* In Section 2.5, describe the electrode layout that was used, as well as the location of the reference electrode.

* In Section 2.7, describe the kernel used in the SVM classifier, as well as the hyperparameters.

* Provide a Conclusion section at the end of the manuscript

* Discuss in how the findings about the ERP amplitude are in line with the results provided in (Hedges, 2016)

Minor comments:

* Pay attention to the capitalization of words, e.g., "Congestive Cardiac Failure" and "Diabetes Mellitus" should not be capitalized.

* Remove extra space in "80 ±20" and all the instances in the manuscript.

* Use always space between quantity and units (e.g., "66 ms" instead of "66ms")

Reviewer #2: The manuscript presents a novel study investigating animacy categorization in MCI patients vs healthy older adults using univariate and multivariate pattern analysis. Results demonstrated decreased speed of processing in MCI patients and differences in activation patterns across a range of electrodes. While the study is well presented, I have a number of concerns around reliability and generalizability of the results, which prevent me from recommending the manuscript from publication.

The statistical analyses appear to have been performed in a technically sound way. However, the small sample size significantly limits the reliability and generalizability of the classification findings. Several studies have highlighted the dangers of small sample sizes when using brain data for classification (e.g. Hosseini et al., 2020). Based on this limitation, it is difficult to interpret the findings (or their significance).

MOCA and ACE-R results should be presented in the manuscript.

The authors should include information on the number of EEG trials rejected at each stage of processing across each condition and for each group. Without this information, it is difficult to interpret the reliability of findings.

The manuscript is presented in an intelligible fashion. However, the Introduction would benefit from more detailed description of previous literature on visual processing in MCI and why this is important to explore (beyond the fact that differences between MCI and HCs have yet to be investigated). Typically, the final paragraph of the Introduction would state the study hypotheses – not the main findings. I suggest the authors revise this accordingly.

Similarly, the Discussion requires a more detailed interpretation of the study findings, especially with regard to the overall pattern of differences found, e.g. why were there group differences in midline frontal, left frontal, left parietal and right centro-parietal electrodes? How does this relate to previous findings?

In my opinion, the authors’ conclusion that the results “suggest the IPS as a potential biomarker for the early detection of AD” is overstated. As mentioned in the Introduction, not all cases of MCI will progress to AD. This fact, coupled with the small patient sample (n=18), significantly limit generalizability to AD.

The authors state that the ICA task is “more challenging than the typical memory tasks, making it more sensitive to less severe brain deteriorations”. This is also at odds with the suggestion of IPS as a potential biomarker for AD, as the task may not be suitable for AD patients. The authors should comment on this in their Discussion.

As far as I can tell, the data underlying the findings have not been made available.

6. PLOS authors have the option to publish the peer review history of their article (what does this mean?). If published, this will include your full peer review and any attached files.

Reviewer #1: No

Reviewer #2: No

---

## [Author Response · Author response to Decision Letter 0]

3 Aug 2021

We would like to thank the Editor for pointing out constructive suggestions. Following these suggestions, based on a power of 0.8, we estimated the sample size to be 22 (for details, please see our response to comment 1 of Reviewer #2). We have also added a paragraph to the Discussion explaining the study limitations (page 18, line 35 to page 19, line 2). 

We would also like to thank the reviewers for pointing out thoughtful comments and great suggestions. We’ve accommodated the comments, and below you can see point-by-point responses to these comments in blue. 

Reviewer #1: Overall:

The manuscript analyzes the ERP response during the integrated cognitive assessment to find differences between healthy controls and patients with mild cognitive impairment.

Results suggest a slower response in the MCI population.

Recommend:

Major comments:

1. In Section 1, relevant works on the use of EEG for AD and MCI assessment are missing, e.g.:

- Regarding resting state EEG: "Systematic Review on Resting-State EEG for Alzheimer’s Disease Diagnosis and Progression Assessment Resting-state EEG" (Cassani, 2018), and

- Regarding ERPs: "P300 Amplitude in Alzheimer’s Disease: A Meta-Analysis and Meta-Regression" (Hedges, 2016)

- Regarding slower EEG signals: "Slowing and Loss of Complexity in Alzheimer's EEG: Two Sides of the Same Coin?" (Dauwels, 2011)

We have added all the above literature to our introduction (page 2, line 9).

2. The MoCA and ACE-R tests are described in Sections 2.2 and 2.3 respectively. In Section 2.4, it is stated the participants completed these tests, however the scores are not presented nor used.

The test results were employed by the consultant neurologist to help with the diagnosis. We added this explanation to the Methods (page 4, line 7, and page 4, line 13). Additionally, the results of both MoCA and ACE are now included in the supplementary materials (Table S1).

3. In Section 2.5, describe the electrode layout that was used, as well as the location of the reference electrode.

This is now added to the supplementary materials (Figure S1). 

4. In Section 2.7, describe the kernel used in the SVM classifier, as well as the hyperparameters.

We added the following subsection to the Methods (section 2.7; page 7 lines 6 to 10), explaining the details of the SVM classifier:

“We used a linear support vector machine (SVM) classifier throughout the analyses to decode neural data. The hyperparameters were as follows: The cost/regularization parameter (C) and the weight of classes were all set to 1. All the classifications were done using the LIBSVM software implementation.“

5. Provide a Conclusion section at the end of the manuscript

We added a section for the conclusion (page 19, Line 4 to 10).

6. Discuss in how the findings about the ERP amplitude are in line with the results provided in (Hedges, 2016)

We added the following to the discussion (page 17, lines 7 to 12), comparing the findings of the current study with that of Hedges, 2016:

“Having a closer look at the ERP of different regions, we observed a decrease in the P300 amplitude of the patients with MCI. This decrease is significant in the electrodes of the midline frontal, right fronto-central, and right temporal regions between 250 ms to 500 ms after the stimulus onset. This observation is in line with the reported result in (Hedges, 2016), especially with the P300 of the midline frontal electrodes, and highlights the importance of this signal in the detection of MCI.”

Minor comments:

* Pay attention to the capitalization of words, e.g., "Congestive Cardiac Failure" and "Diabetes Mellitus" should not be capitalized. 

 Done

* Remove extra space in "80 ±20" and all the instances in the manuscript. 

 Done

* Use always space between quantity and units (e.g., "66 ms" instead of "66ms") 

 Done

Reviewer #2: The manuscript presents a novel study investigating animacy categorization in MCI patients vs healthy older adults using univariate and multivariate pattern analysis. Results demonstrated decreased speed of processing in MCI patients and differences in activation patterns across a range of electrodes. While the study is well presented, I have a number of concerns around reliability and generalizability of the results, which prevent me from recommending the manuscript from publication.

1. The statistical analyses appear to have been performed in a technically sound way. However, the small sample size significantly limits the reliability and generalizability of the classification findings. Several studies have highlighted the dangers of small sample sizes when using brain data for classification (e.g. Hosseini et al., 2020). Based on this limitation, it is difficult to interpret the findings (or their significance).

We understand that the number of participants might raise a concern. However, these findings are supported by our previous behavioral results from MCI patients, which indicate a significant decrease in the speed and accuracy of categorizing animacy information (SM Khaligh-Razavi, 2019, Scientific Report , with 448 participants, and MH Modarres, 2021, bioRxiv (accepted at Front Psychiatry) , with 230 participants). The current study was designed for taking a closer look at the underlying neural response of MCI patients and healthy individuals while doing a task that already showed their difference in speed and accuracy at the behavioral level.

Furthermore, the number of individuals who participated in our study is comparable to the recent related EEG publications on MCI/AD. For reference here is a list of such publications: 

1. Zhao et al., 2020, Effects of creative expression program on the event-related potential and task reaction time of elderly with mild cognitive impairment, International journal of nursing sciences

Control MCI: 18, MCI: 18

2. Massa et al., 2020, Utility of quantitative EEG in early Lewy body disease, Parkinsonism & related disorders

 HC: 24, MCI-LBD: 12, MCI-AD: 11

3. Briels et al., 2020, Profound regional spectral, connectivity, and network changes reflect visual deficits in posterior cortical atrophy: an EEG study, Neurobiology of Aging

 HC: 29, PCA (Posterior Cortical Atrophy)-AD: 29

4. Sharma et al., 2019, EEG and cognitive biomarkers based mild cognitive impairment diagnosis, IRBM

 HC: 13, MCI: 16, dementia: 15

5. Fraga et al. 2018, Early diagnosis of mild cognitive impairment and Alzheimer’s with event-related potentials and event-related desynchronization in N-back working memory tasks, Computer methods and programs in biomedicine

 HC: 27, MCI: 21, AD: 15

6. Simons et al., 2018, Fuzzy Entropy Analysis of the Electroencephalogram in Patients with Alzheimer’s Disease: Is the Method Superior to Sample Entropy?, Entropy

 HC: 11, AD: 11

7. Azami et al., 2017, Univariate and Multivariate Generalized Multiscale Entropy to Characterise EEG Signals in Alzheimer’s Disease, Entropy

 HC: 11, AD: 11

8. Han et al., 2017, Changes of EEG Spectra and Functional Connectivity during an Object-Location Memory Task in Alzheimer’s Disease, Frontiers in behavioral neuroscience

 HC: 19, mild AD: 20

9. Fraga et al. 2017, Event-related synchronisation responses to N-back memory tasks discriminate between healthy ageing, mild cognitive impairment, and mild Alzheimer's disease, IEEE International Conference on Acoustics, Speech, and Signal Processing (ICASSP)

 HC: 27, MCI: 21

Following the editor’s suggestion, we also estimated the sample size needed for the desired power of 0.8 (when comparing the decoding peaks of HC vs. MCI). For μ1 = 434 (median of HC peak time points), μ2 = 473 (median of MCI peak time points), σ = 32.28 (pooled standard deviation of HC and MCI peak time points), α = 0.001 (type I error rate), the sample size of each group was estimated as 22, which is also consistent with the mentioned studies.

2. MOCA and ACE-R results should be presented in the manuscript.

We added these data as well as the data for the ICA test to the Supplementary materials (Table S1 of supplementary materials).

3. The authors should include information on the number of EEG trials rejected at each stage of processing across each condition and for each group. Without this information, it is difficult to interpret the reliability of findings.

We included resting trials in between the image trials (i.e., almost every 70 seconds, they were given 10 seconds to rest their eyes, blink, and swallow). Participants were instructed to only blink (or swallow) during these trials to prevent contamination of EEG signals with the eye-blink (and swallowing) artifacts. These trials were excluded from further EEG analyses. Because of such a design, we did not have to reject any of the image trials. Other potential artifacts were removed with Independent Component Analysis. We added this description to the methods (section 2.5, page 6, lines 18 to 24).

4. The manuscript is presented in an intelligible fashion. However, the Introduction would benefit from more detailed description of previous literature on visual processing in MCI and why this is important to explore (beyond the fact that differences between MCI and HCs have yet to be investigated). Typically, the final paragraph of the Introduction would state the study hypotheses – not the main findings. I suggest the authors revise this accordingly.

We updated the introduction to include more information about the importance of the visual system and the ICA task with regard to cognitive impairment in MCI/AD. (See page 2, line 27-29 and line 33 of page 2 to line 7 of page 3).

There are various styles in scientific writing. We favor that of Mensh & Kording, Plos Comp (2017) ; this suggests the inclusion of the main findings’ summary. Following the reviewer’s suggestion, we also added a few sentences on study hypotheses.

5. Similarly, the Discussion requires a more detailed interpretation of the study findings, especially with regard to the overall pattern of differences found, e.g. why were there group differences in midline frontal, left frontal, left parietal and right centro-parietal electrodes? How does this relate to previous findings?

We added the sub-section (4.2) to the Discussion on how these changes relate to previous findings (page 18, line 1 to line 16).

6. In my opinion, the authors’ conclusion that the results “suggest the IPS as a potential biomarker for the early detection of AD” is overstated. As mentioned in the Introduction, not all cases of MCI will progress to AD. This fact, coupled with the small patient sample (n=18), significantly limit generalizability to AD.

We have modified the discussion to mention the study limitations (page 18 line 35 to page 19, line 2).

It is worth noting (as also added to the discussion; page 18 lines 32 to 34) that the purpose of this study was to detect mild cognitive impairment. We do not necessarily claim that all MCI cases are due to AD or will convert to AD. There is clinical value in detecting MCI patients via large-scale population screening using digital biomarkers, such as ICA. The next step is then to use more sophisticated tools, such as PET, CSF, etc to identify those who are more likely to develop AD or have an amyloid burden.

7. The authors state that the ICA task is “more challenging than the typical memory tasks, making it more sensitive to less severe brain deteriorations”. This is also at odds with the suggestion of IPS as a potential biomarker for AD, as the task may not be suitable for AD patients. The authors should comment on this in their Discussion.

We added this to the discussion (page 16 lines 28 to 33):

The ICA is particularly suited for population-wide screening of cognitive impairment to help identify patients with MCI and mild AD (MiAD) and not designed for cognitive assessment in more severe stages of the disease, such as moderate to severe AD. Early detection of cognitive impairment is becoming increasingly important, particularly in the context of disease-modifying therapies for early stages of AD, such as Aducanumab –which has recently received FDA approval. 

8. As far as I can tell, the data underlying the findings have not been made available.

The data for the behavioral tests (MoCA, ACE-R, and ICA) can be found in the supplementary materials (S1 Table). The EEG dataset related to the findings in the presented manuscript is available at RepOD (https://doi.org/10.18150/DEQMGF).

---

## [Decision Letter · Decision Letter 1]

3 Feb 2022

Temporal dynamics of animacy categorization in the brain of patients with mild cognitive impairment

PONE-D-21-04261R1

Dear Dr. Karimi,

We’re pleased to inform you that your manuscript has been judged scientifically suitable for publication and will be formally accepted for publication once it meets all outstanding technical requirements.

Kind regards,

Stavros I. Dimitriadis

Academic Editor

PLOS ONE

Additional Editor Comments (optional):

The authors responded properly to the reviewers' comments.

I have reviewed personally the draft and I have no further comments to add.

I recommend the acceptance of the manuscript.

Reviewers' comments:

Reviewer's Responses to Questions

**Comments to the Author**

1. If the authors have adequately addressed your comments raised in a previous round of review and you feel that this manuscript is now acceptable for publication, you may indicate that here to bypass the “Comments to the Author” section, enter your conflict of interest statement in the “Confidential to Editor” section, and submit your "Accept" recommendation.

Reviewer #2: All comments have been addressed

2. Is the manuscript technically sound, and do the data support the conclusions?

Reviewer #2: Yes

3. Has the statistical analysis been performed appropriately and rigorously? 

Reviewer #2: Yes

4. Have the authors made all data underlying the findings in their manuscript fully available?

Reviewer #2: Yes

5. Is the manuscript presented in an intelligible fashion and written in standard English?

Reviewer #2: Yes

6. Review Comments to the Author

Reviewer #2: All comments have been addressed. I have no further suggestions for revisions.

7. PLOS authors have the option to publish the peer review history of their article (what does this mean?). If published, this will include your full peer review and any attached files.

Reviewer #2: No

---

## [Editor Report · Acceptance letter]

9 Feb 2022

PONE-D-21-04261R1 

Temporal dynamics of animacy categorization in the brain of patients with mild cognitive impairment 

Dear Dr. Karimi:

I'm pleased to inform you that your manuscript has been deemed suitable for publication in PLOS ONE. Congratulations! Your manuscript is now with our production department. 

Kind regards, 

on behalf of

Dr. Stavros I. Dimitriadis 

Academic Editor

PLOS ONE